# *Sox8* and *Sox9* act redundantly for ovarian-to-testicular fate reprogramming in the absence of *R-spondin1* in mouse sex reversals

Nainoa Richardson[1], Isabelle Gillot[1], Elodie P Gregoire[1], Sameh A Youssef[2,3], Dirk de Rooij[4], Alain de Bruin[2,3], Marie-Cécile De Cian[1†], Marie-Christine Chaboissier[1†*]

[1]Université Côte d'Azur, CNRS, Inserm, iBV, Nice, France; [2]Department of Pathobiology, Dutch Molecular Pathology Center, Faculty of Veterinary Medicine, Utrecht University, Utrecht, Netherlands; [3]Department Pediatrics, Divisions Molecular Genetics, University Medical Center Groningen, University of Groningen, Groningen, Netherlands; [4]Department of Biology, Faculty of Science, Division of Developmental Biology, Reproductive Biology Group, Utrecht University, Utrecht, Netherlands

**\*For correspondence:**
chaboiss@unice.fr

[†]These authors contributed equally to this work

**Competing interests:** The authors declare that no competing interests exist.

**Abstract** In mammals, testicular differentiation is initiated by transcription factors SRY and SOX9 in XY gonads, and ovarian differentiation involves R-spondin1 (RSPO1) mediated activation of WNT/β-catenin signaling in XX gonads. Accordingly, the absence of *RSPO1/Rspo1* in XX humans and mice leads to testicular differentiation and female-to-male sex reversal in a manner that does not require*Sry* or *Sox9* in mice. Here we show that an alternate testis-differentiating factor exists and that this factor is *Sox8*. Specifically, genetic ablation of *Sox8* and *Sox9* prevents ovarian-to-testicular reprogramming observed in XX *Rspo1* loss-of-function mice. Consequently, *Rspo1 Sox8 Sox9* triple mutant gonads developed as atrophied ovaries. Thus, SOX8 alone can compensate for the loss of SOX9 for Sertoli cell differentiation during female-to-male sex reversal.

## Introduction

During primary sex determination in mammals, a common precursor organ, the bipotential gonad, develops as a testis or ovary. In humans and mice, testicular development begins when SRY and SOX9 are expressed in the bipotential XY gonad. These transcription factors promote supporting cell progenitors to differentiate as Sertoli cells and form sex cords (*Gonen et al., 2018*; *Chaboissier et al., 2004*; *Barrionuevo et al., 2006*), and this triggers a cascade of signaling events that are required for the differentiation of other cell populations in the testis (*Koopman et al., 1991*; *Vidal et al., 2001*). In XX embryos, the bipotential gonad differentiates as an ovary through a process that requires RSPO1-mediated activation of canonical WNT/β-catenin (CTNNB1) signaling in somatic cells (*Parma et al., 2006*; *Chassot et al., 2008*). Ovarian fate also involves activation of FOXL2, a transcription factor that is required in post-natal granulosa cells (*Schmidt et al., 2004*; *Ottolenghi et al., 2005*; *Uhlenhaut et al., 2009*), which organize as follicles during embryogenesis in humans and after birth in mice (*McGee and Hsueh, 2000*; *Mork et al., 2012*). For complete differentiation of testes or ovaries, an active repression of the opposite fate is necessary (*Kim et al., 2006*). Inappropriate regulation within the molecular pathways governing sex determination can lead to partial or complete sex reversal phenotypes and infertility (*Wilhelm et al., 2009*).

**eLife digest** In humans, mice and other mammals, genetic sex is determined by the combination of sex chromosomes that each individual inherits. Individuals with two X chromosomes (XX) are said to be chromosomally female, while individuals with one X and one Y chromosome (XY) are chromosomally males.

One of the major differences between XX and XY individuals is that they have different types of gonads (the organs that make egg cells or sperm). In mice, for example, before males are born, a gene called *Sox9* triggers a cascade of events that result in the gonads developing into testes. In females, on the other hand, another gene called *Rspo1* stimulates the gonads to develop into ovaries.

Loss of *Sox9* in XY embryos, or *Rspo1* in XX embryos, leads to mice developing physical characteristics that do not match their genetic sex, a phenomenon known as sex reversal. For example, in XX female mice lacking *Rspo1,* cells in the gonads reprogram into testis cells known as Sertoli cells just before birth and form male structures known as testis cords. The gonads of female mice missing both *Sox9* and *Rspo1* (referred to as "double mutants") also develop Sertoli cells and testis cords, suggesting another gene may compensate for the loss of *Sox9*.

Previous studies suggest that a gene known as *Sox8*, which is closely related to *Sox9*, may be able to drive sex reversal in female mice. However, it was not clear whether *Sox8* is able to stimulate testis to form in female mice in the absence of *Sox9*.

To address this question, Richardson et al. studied mutant female mice lacking *Rspo1*, *Sox8* and *Sox9*, known as "triple mutants". Just before birth, the gonads in the triple mutant mice showed some characteristics of sex reversal but lacked the Sertoli cells found in the double mutant mice. After the mice were born, the gonads of the triple mutant mice developed as rudimentary ovaries without testis cords, unlike the more testis-like gonads found in the double mutant mice.

The findings of Richardson et al. show that *Sox8* is able to trigger sex reversal in female mice in the absence of *Rspo1* and *Sox9*. Differences in sexual development in humans affect the appearance of individuals and often cause infertility. Identifying *Sox8* and other similar genes in mice may one day help to diagnose people with such conditions and lead to the development of new therapies.

Studies in humans and mice have shown that the pathway initiated by SRY/SOX9 or RSPO1/WNT/β-catenin signaling are indispensable for sex specific differentiation of the gonads. For example, in XY humans, *SRY* or *SOX9* loss-of-function mutations prevent testis development (**Berta et al., 1990**; **Houston et al., 1983**). In mice, XY gonads developing without SRY or SOX9 lack Sertoli cells and seminiferous tubules and differentiate as ovaries that contain follicles (**Lovell-Badge and Robertson, 1990**; **Chaboissier et al., 2004**; **Barrionuevo et al., 2006**; **Lavery et al., 2011**; **Kato et al., 2013**), indicating *Sry/Sox9* requirement. In XX humans and mice, *SRY/Sry* or *SOX9/Sox9* gain-of-function mutations promote Sertoli cell differentiation and testicular development (**Sinclair et al., 1990**; **Koopman et al., 1991**; **Bishop et al., 2000**; **Vidal et al., 2001**; **Huang et al., 1999**), indicating that SRY/SOX9 function is also sufficient for male gonad differentiation.

With respect to the ovarian pathway, homozygous loss-of-function mutations for *RSPO1/Rspo1* trigger partial female-to-male sex reversal in XX humans and mice (**Parma et al., 2006**; **Chassot et al., 2008**). In XX *Rspo1* or *Wnt4* mutant mice, Sertoli cells arise from a population of embryonic granulosa cells (pre-granulosa cells) that precociously exit their quiescent state, differentiate as mature granulosa cells, and reprogram as Sertoli cells (**Chassot et al., 2008**; **Maatouk et al., 2013**). The resulting gonad is an ovotestis containing seminiferous tubule-like structures with Sertoli cells and ovarian follicles with granulosa cells, indicating that SRY is dispensable for testicular differentiation. In addition, stabilization of WNT/CTNNB1 signaling in XY gonads leads to male-to-female sex reversal (**Maatouk et al., 2008**; **Harris et al., 2018**). Thus, RSPO1/WNT/CTNNB1 signaling is required for ovarian differentiation and female development in humans and mice.

Given the prominent role of SOX9 for testicular development (**Chaboissier et al., 2004**; **Barrionuevo et al., 2009**), it was hypothesized that SOX9 is responsible for Sertoli cell differentiation in XX gonads developing without *RSPO1/Rspo1*. This hypothesis was tested by co-inactivation of *Rspo1* or *Ctnnb1* and *Sox9* in *Rspo1$^{-/-}$; Sox9$^{fl/fl}$; Sf1:cre$^{Tg/+}$* (**Lavery et al., 2012**) and in *Ctnnb1$^{fl/fl}$;*

Sox9$^{fl/fl}$; Sf1:cre$^{Tg/+}$double mutant mice (**Nicol and Yao, 2015**). Unexpectedly, XY and XX Rspo1 or Ctnnb1 mutant gonads lacking Sox9 exhibited Sertoli cells organized as testis cords (**Nicol and Yao, 2015**; **Lavery et al., 2012**). Specifically, gonads in XX Rspo1$^{-/-}$; Sox9$^{fl/fl}$; Sf1:cre$^{Tg/+}$ double mutant mice developed as ovotestes as in XX Rspo1$^{-/-}$ single mutants, and XY Rspo1$^{-/-}$; Sox9$^{fl/fl}$; Sf1:cre$^{Tg/+}$ mutant mice developed hypoplastic testes capable of supporting the initial stages of spermatogenesis. These outcomes indicate that at least one alternate factor can promote testicular differentiation in Rspo1 mutant mice also lacking Sox9 in XY mice, and lacking both Sry and Sox9 in XX animals. This or these factors remained to be identified.

Among the candidate genes that could promote testicular differentiation in the absence of Sry and Sox9 are the other members of the SoxE group of transcription factors that includes Sox9, Sox8 and Sox10 (**Lavery et al., 2012**; **Nicol and Yao, 2015**). However, Sox10 expression in testes depends on Sox8 and Sox9 (**Georg et al., 2012**), and Sox10 loss-of-function mice are fertile (**Britsch et al., 2001**; **Peirano and Wegner, 2000**), suggesting that Sox10 would not be the best candidate gene. For Sox8, loss-of-function analyses in XY gonads show testicular development, indicating that Sox8 is not required for Sertoli cell differentiation during embryonic development (**Sock et al., 2001**). However, a Sox8-null background enhanced the penetrance of the testis-to-ovary sex reversal phenotype in mice with reduced Sox9 expression (**Chaboissier et al., 2004**), suggesting that Sox8 supports the function of Sox9.

Furthermore, in XY Sox9$^{fl/fl}$; Sf1:cre$^{Tg/+}$ single mutant mice and in XY Sox8$^{-/-}$; Sox9$^{fl/fl}$; Amh:cre$^{Tg/+}$ and Sox8$^{-/-}$; Sox9$^{fl/fl}$; Wt1-CreER$^{T2/+}$ double mutant mice where Sox9 is inactivated after sex determination, the single and double mutant mice initially form testis cords containing Sertoli cells. However, these cells then lose their identity and begin to express granulosa cell markers like FOXL2 (**Barrionuevo et al., 2009**; **Barrionuevo et al., 2016**; **Georg et al., 2012**). In addition, following tamoxifen induction of Cre recombinase and subsequent deletion of Sox9, Sertoli cells in Sox8$^{-/-}$; Sox9$^{fl/fl}$; Wt1-CreER$^{T2/+}$ testes become apoptotic leading to a complete degeneration of the seminiferous tubules. This indicated that a concerted effort by Sox8 and Sox9 is required in XY gonads for the maintenance of Sertoli cells after sex determination. Beyond mice, in humans, SOX8 contributes to testis differentiation or homeostasis, given the 46,XY gonadal dysgenesis phenotype associated with mutations/rearrangements at the SOX8 locus (**Portnoi et al., 2018**).

Although Sox8 expression is dispensable for Sertoli cell differentiation in XY gonads, it may have a key role for testicular differentiation in XX sex reversal gonads or in cases of Sox9-independent testicular differentiation in XY gonads. This led us to hypothesize that Sox8 can compensate for loss of Sox9 and is the alternate factor capable of: (*i*) triggering sex reversal in XX Rspo1 knockout gonads lacking Sry and Sox9, and (*ii*) promoting testicular development in XY Rspo1 knockout gonads lacking Sox9.

To test this hypothesis, we have generated triple Rspo1, Sox8, and Sox9 loss-of-function mutant mice models. We show here that Sox8 and Sox9 are individually dispensable for testicular development in XY and XX mice lacking Rspo1, indicating the presence of redundant testicular pathways. In the absence of both Sox factors, Sertoli cell differentiation is precluded and XY and XX Rspo1$^{-/-}$; Sox8$^{-/-}$; Sox9$^{fl/fl}$; Sf1:cre$^{Tg/+}$ triple mutants develop atrophied ovaries. Together, our data show that Sox8 or Sox9 is required to induce testicular development in XY and XX mice lacking Rspo1.

## Results

### Rspo1, Sox8, and Sox9 are expressed independently

We first performed expression analyses for Rspo1 (**Figure 1A a-h**), Sox8 (**Figure 1 Ba-l**), and Sox9 (**Figure 1 Ca-l**), in control and mutant gonads. We chose to study embryonic day 17.5 (E17.5) fetal gonads, when testis cords form in Rspo1 sex reversal mice, and juvenile postnatal day 10 (P10) gonads, when gonadal fate is likely to be set (**Lavery et al., 2012**). In XY gonads, Rspo1 is mostly localized to the coelomic epithelium at E17.5 and to the tunica albuginea at P10 (**Figure 1 Aa, c**). In fetal ovaries, Rspo1 is expressed in somatic cells at E17.5 and down-regulated after birth, as shown in post-natal P10 ovaries (**Figure 1 Af,h**). In XY and XX mice lacking Sox8 and Sox9 (i.e., Sox8$^{-/-}$; Sox9$^{fl/fl}$; Sf1:cre$^{Tg/+}$, referred to as Sox8$^{KO}$ Sox9$^{cKO}$ double mutants), high Rspo1 expression levels were observed in embryonic gonads and down-regulated after birth, indicating ovarian differentiation (**Figure 1 Ab,d,e,g**), as previously described (**Chaboissier et al., 2004**). Together, these data

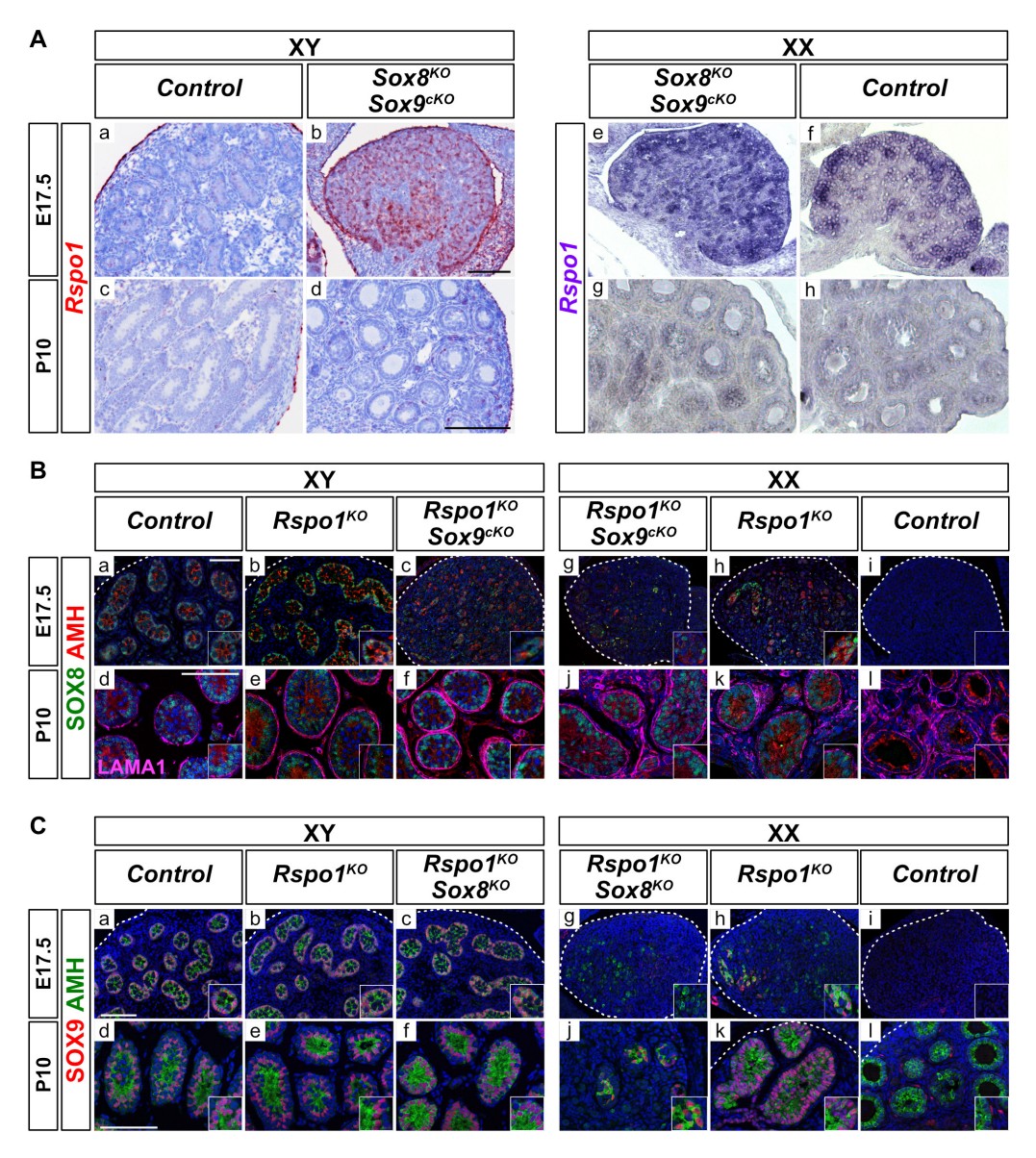

**Figure 1.** Expression of *Rspo1*, *Sox8*, and *Sox9* in E17.5 and P10 gonads. Expression of *Rspo1*, as revealed by in situ hybridizations (**A**), and of SOX8 and SOX9, as revealed by immunostaining (**B, C**) on gonadal sections from embryonic day 17.5 (E17.5) and 10 days post-natal (P10) mice. In XY wild-type testes, *Rspo1* is mainly expressed in the coelomic epithelium (**Aa**) and tunica albuginea (**Ac**). In XX wildtype ovaries, *Rspo1* is expressed throughout the gonad at E17.5 (**Af**), and down-regulated in post-natal animals, as shown at P10 (**Ah**). In XY and XX *Sox8*[-/-]; *Sox9*[flox/flox]; *Sf1:cre*[Tg/+] (*Sox8*[KO] *Sox9*[cKO]) mutant mice, the *Rspo1* expression profile (**Ab,d,e,g**) is similar to wildtype ovaries (**Af,h**), indicating an ovarian fate. For SOX8 expression, in XY control testes (**Ba,d**) and in XY *Rspo1*[KO] gonads developing as testes (**Bb,e**), SOX8 is expressed in testis cords at E17.5 and P10. Co-immunolabeling with AMH confirmed the identity of Sertoli cells. In XX mice, though AMH is expressed in post-natal control ovaries, these cells were SOX8-negative, indicating that they are granulosa cells (**Bj**). However, SOX8 and AMH positive testis cords were found in XX *Rspo1*[KO] female-to-male sex reversal gonads (**Bh,k**). SOX8 is also expressed in the absence of *Sox9* in XY and XX *Rspo1*[-/-]; *Sox9*[flox/flox]; *Sf1:cre*[Tg/+] (*Rspo1*[KO] *Sox9*[cKO]) gonads at E17.5 and P10 (**Bc,g,f,j**). At P10, LAMA1 staining demarcates testis cords (**Bd-f, j–k**) and follicles (**Bl**). SOX9 expression was found in XY control testes (**Ca,d**), and in XY *Rspo1*[KO] gonads developing as testes (**Cb,e**). Co-immunolabeling with AMH confirmed the identity of Sertoli cells. As shown, SOX9 and AMH positive testis cords are found in XX *Rspo1*[KO] sex reversal gonads at E17.5 and P10 (**Ch,k**). In addition, SOX9 is also expressed in absence of *Sox8* in XY Rspo1-/-; *Sox8*[-/-] (*Rspo1*[KO] *Sox8*[KO]) gonads developing as testes at E17.5 and P10 (**Cc,f**), and in XX *Rspo1*[KO] *Sox8*[KO] gonads developing as ovotestes at P10 (**Cd**). In XX control mice, SOX9 and AMH expression is absent in fetal ovaries (**Ci**). In post-natal female animals, SOX9 is expressed by theca cells, which are AMH-negative (**Cj**). All scale bars 100 µm.

confirmed that although *Rspo1* is expressed in both XY and XX gonads, robust *Rspo1* expression in cells throughout the gonad is a feature of ovarian development in fetuses.

In XY control, XY *Rspo1$^{-/-}$* (referred to as *Rspo1$^{KO}$*), and XY *Rspo1$^{-/-}$; Sox9$^{fl/fl}$; Sf1:cre$^{Tg/+}$* (referred to as *Rspo1$^{KO}$ Sox9$^{cKO}$*) mice, immunostaining revealed SOX8 expression in Sertoli cells organized as testis cords at E17.5 and seminiferous tubules at P10, in agreement with previous reports (*Figure 1 Ba,d,b,e,g,j*; *Schepers et al., 2003*; *Lavery et al., 2012*). In XX mice, though SOX8 is not expressed in control ovaries (*Figure 1 Bi,l*), expression was observed in XX *Rspo1$^{KO}$* and XX *Rspo1$^{KO}$ Sox9$^{cKO}$* sex reversal gonads (*Figure 1 Bh,k,g,j*). Co-immunolabeling with AMH confirmed the identity of Sertoli cells (*Figure 1 Ba-f, g-h, j-l*) and LAMA1 staining at P10 demarcated both testis cords (*Figure 1 Bd-f, j-k*) and follicles (*Figure 1l*), which do not express SOX8. In summary, these data corroborated that *Sox8* is expressed in gonads lacking *Rspo1*, and that its expression can be independent of *Sox9* (*Lavery et al., 2012*).

Next, immunostaining revealed SOX9-positive testis cords in XY *Rspo1$^{KO}$* testes (*Figure 1 Cb,e*), XX *Rspo1$^{KO}$* ovotestes (*Figure 1 Ch, k*), as in control testes (*Figure 1 Ca, d*), as previously described (*Chassot et al., 2008*). Co-immunolabeling with AMH confirmed the identity of Sertoli cells, since AMH-positive granulosa cells do not express *Sox9*, and given that ovarian steroidogenic theca cells expressing *Sox9* are AMH-negative (*Figure 1Cl*). In addition, deletion of *Sox8* did not alter the expression of *Sox9* in XY or XX *Rspo1$^{KO}$* gonads (i.e., in *Rspo1$^{KO}$ Sox8$^{KO}$* gonads) (*Figure 1Cc,f,g,j*). Altogether, our results show that *Sox8* and *Sox9* are expressed in the absence of each other in *Rspo1* mutant gonads when testis cords are present or when partial sex reversal occurs.

## Ablation of *Rspo1* and *Sox8* does not impair testis differentiation

Next, we asked how inactivation of both *Rspo1* and *Sox8* would impact gonad development in XY and XX *Rspo1$^{KO}$ Sox8$^{KO}$* double mutants by comparison with controls (*Figure 2a–y* and *Figure 2—figure supplement 1a–h*). In XY *Rspo1$^{KO}$ Sox8$^{KO}$* mice, the anogenital distance in adult P40 animals was comparable to XY control males (*Figure 2a,b*). In contrast, XX control females exhibited a short anogenital distance (*Figure 2m*). Internally, XY *Rspo1$^{KO}$ Sox8$^{KO}$* mice developed epididymides, vasa deferensia, seminal vesicles and prostate, as in control males (*Figure 2—figure supplement 1a,b*). Histological analyses by PAS staining revealed seminiferous tubules with no obvious defects in P10 and P40 XY *Rspo1$^{KO}$ Sox8$^{KO}$* animals (*Figure 2—figure supplement 1c,d* and *Figure 2e,f*), and these mice were fertile. Testicular development in XY *Rspo1$^{KO}$ Sox8$^{KO}$* mice was confirmed by immunostaining experiments on embryonic (E17.5) and post-natal (P10, and P40) gonads that contained SOX9 and DMRT1 positive Sertoli cells forming testicular sex cords and seminiferous tubules (*Figure 1Cc,f*, *Figure 2g–j*, and *Figure 2—figure supplement 1e–h*). DMRT1 expression was also observed in germ cells, which are TRA98-positive (*Figure 2i,j* and *Figure 2—figure supplement 1g, h*; *Matson et al., 2010*). Thus, loss of both *Rspo1* and *Sox8* does not impair testis differentiation.

For XX *Rspo1$^{KO}$ Sox8$^{KO}$* mice, the question is whether the double mutant gonads developed as ovaries or as ovotestes, as in XX *Rspo1$^{KO}$* single mutant (*Figure 2k–y* and *Figure 2—figure supplement 1i–t*) and as in XX *Rspo1$^{KO}$ Sox9$^{cKO}$* double mutant mice (*Lavery et al., 2012*). Externally, as in XX control mice, both XX *Rspo1$^{KO}$* and XX *Rspo1$^{KO}$ Sox8$^{KO}$* mice developed a short anogenital distance, as shown in adult P40 animals (*Figure 2k–m*). Internally, although XX *Rspo1$^{KO}$ Sox8$^{KO}$* mice exhibited rare testis cords during embryonic development (*Figure 1Cg*), seminiferous tubules devoid of germ cells were apparent at P10, suggesting a delay in ovo-testicular development in double mutant gonads (*Figure 2—figure supplement 1l,m*). Indeed, by P40, both XX *Rspo1$^{KO}$* and XX *Rspo1$^{KO}$ Sox8$^{KO}$* mice were essentially indistinguishable with respect to gonad morphology (*Figure 2n,o*), reproductive tract development (*Figure 2—figure supplement 1i,j*), ovo-testicular organization (*Figure 2q,r*), and the presence of SOX9 and DMRT1 positive Sertoli cells in the testicular area (*Figure 2t,u,w,x*). Altogether, studies performed in *Rspo1$^{KO}$ Sox8$^{KO}$* mice demonstrate that like *Sox9* (*Lavery et al., 2012*), *Sox8* is dispensable for testicular development in XY and XX *Rspo1$^{KO}$* gonads. Moreover, our data suggests that SOX9 likely compensates for the loss of *Sox8* in *Rspo1$^{KO}$ Sox8$^{KO}$* double mutants.

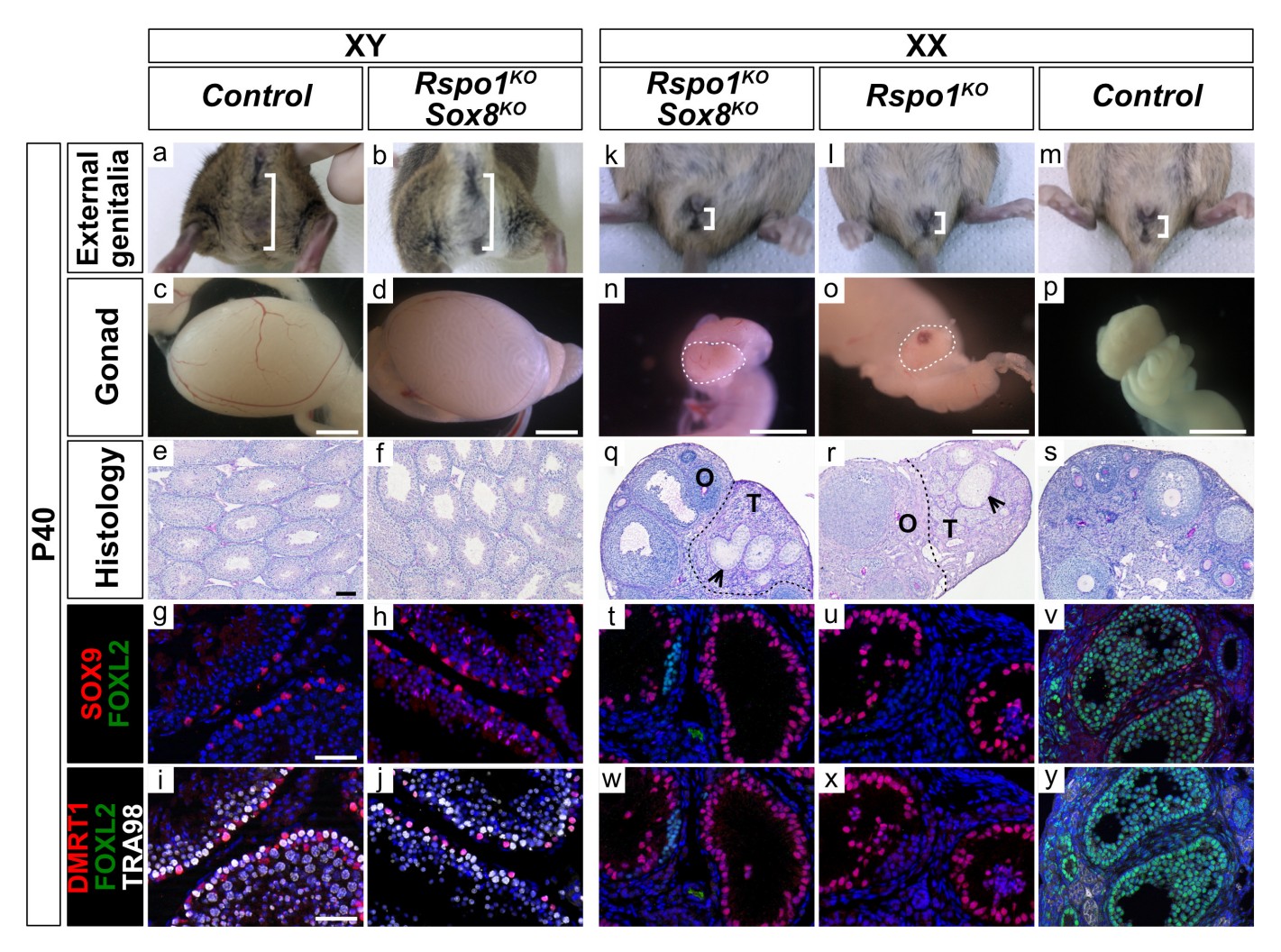

**Figure 2.** External genitalia and gonad development in adult XY and XX *Rspo1^KO Sox8^KO* double mutant mice. External genitalia from adult P40 mice (a–b, k–m), macroscopic view of gonads (c–d, n–p) (Scale bars 1.5 mm), histology as revealed by PAS staining on gonadal sections (e–f, q–s) (Scale bars 100 µm), and immunostaining of SOX9 (Sertoli cell marker, in red) (g–h, t–v), FOXL2 (granulosa cell marker, in green) (g–j, t–y), DMRT1 (Sertoli and germ cell marker, in red) (i–j, w–y), TRA98 (germ cell marker, in white) (i–j, w–y) and DAPI (nuclear marker, in blue) (g–j, t–y) on gonadal sections (Scale bars 50 µm). Inactivation of both *Rspo1* and *Sox8* in XY *Rspo1^KO Sox8^KO* double mutant mice did not cause a sex reversal (a–j). XY *Rspo1^KO Sox8^KO* gonads developed as testes with seminiferous tubules (f) containing SOX9 and DMRT1 positive Sertoli cells (h–j), as in control testes (e, g, i). As shown, XX control ovaries developed follicles (s) containing FOXL2-positive granulosa cells (v, y). Adult ovotestes in XX *Rspo1^KO Sox8^KO* mice (n, q) were indistinguishable from XX *Rspo1^KO* mice (o, r). These gonads contained an ovarian 'O' compartment with follicles and a testicular 'T' compartment with seminiferous tubule-like structures, as indicated by arrowheads (q, r). The seminiferous tubule-like structures in XX *Rspo1^KO Sox8^KO* and XX *Rspo1^KO* ovotestes contained SOX9 and DMRT1 positive Sertoli cells (t–u, w–x), as in control testes (g, i), but lacked TRA98-positive germ cells (w, x).
The online version of this article includes the following figure supplement(s) for figure 2:

**Figure supplement 1.** Reproductive tract of XY and XX *Rspo1^KO Sox8^KO* adult P40 mice and analyses in juvenile P10 mice.

## Ovarian precocious differentiation occurs in XX and XY *Rspo1^KO Sox8^KO Sox9^cKO* fetuses

Our genetic mouse models allowed us to investigate gonadal fate in XY and XX *Rspo1^KO* mice lacking both *Sox8* and *Sox9* (i.e., in XY and XX *Rspo1^KO Sox8^KO Sox9^cKO* triple mutant mice). We first studied gonads in E17.5 fetuses (*Figure 3a–j''* and *Figure 3—figure supplement 1a–j'*), which is when differentiated granulosa cells reprogram as Sertoli cells in XX *Rspo1^KO* gonads (*Maatouk et al., 2013*). As shown, XX control gonads contained granulosa cells expressing FOXL2, but not Sertoli cells expressing SOX9 or DMRT1 (*Figure 3f* and *Figure 3—figure supplement 1a*),

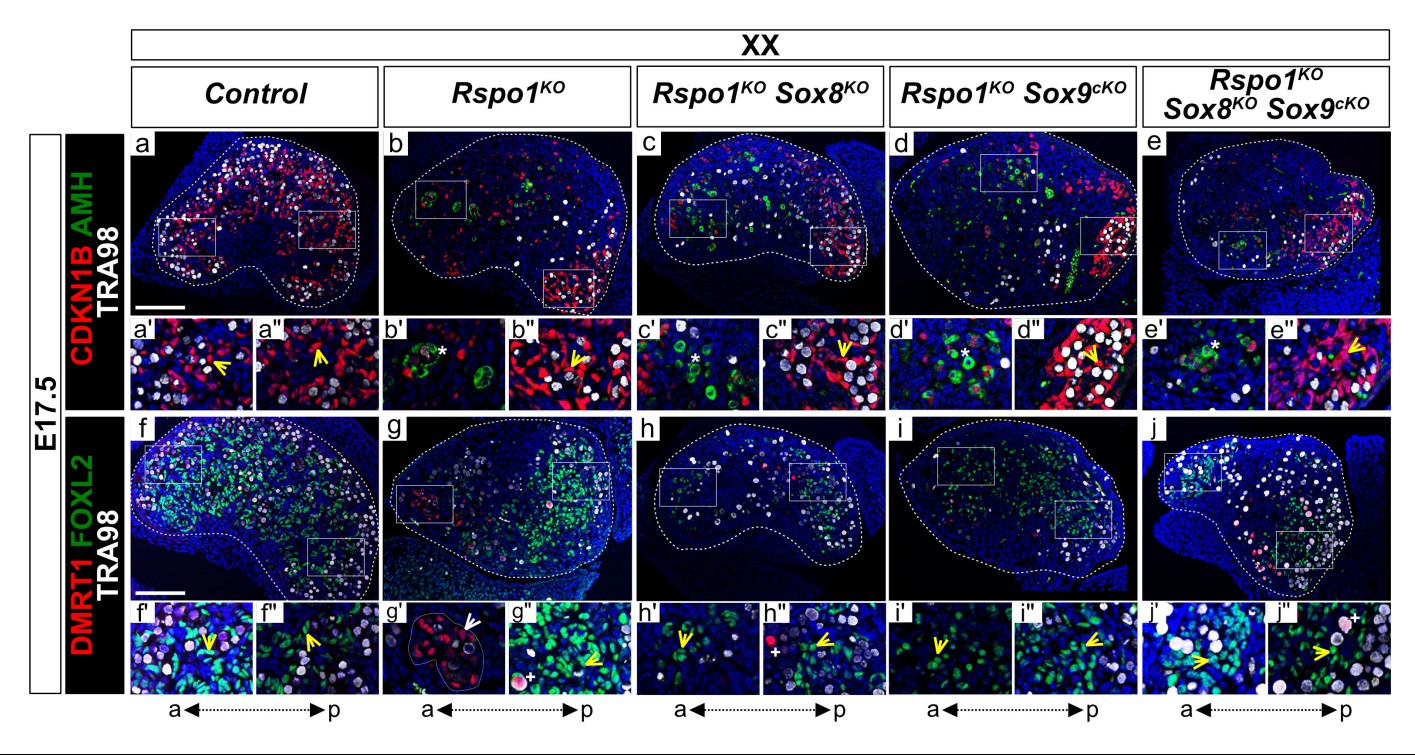

**Figure 3.** Precocious granulosa cell differentiation in XX *Rspo1^KO^ Sox8^KO^ Sox9^cKO^* triple mutant fetuses at E17.5. Immunofluorescence of CDKN1B (P27) (mitotic arrest marker, in red) (a–e''), AMH (Sertoli marker and mature granulosa cell marker, in green) (a–e''), DMRT1 (Sertoli and germ cell marker, in red) (f–j''), FOXL2 (granulosa cell marker, in green) (f–j''), TRA98 (germ cell marker, in white) (a–j''), and DAPI (nuclear marker, in blue) (a–j'') on gonadal sections from E17.5 fetuses (main panels scale bar 100 μm). The anterior 'a' and posterior 'p' axis is shown below each column. For main panels (a–j), highlighted anterior and posterior areas are shown in the respective single and double primed letter panels. Yellow arrowheads indicate granulosa cells expressing CDKN1B or FOXL2, asterisks indicate cells expressing AMH, white arrowheads indicate Sertoli cells expressing DMRT1, and plus symbols indicate germ cells expressing DMRT1 and TRA98. Gonads in XX control fetuses developed as ovaries, as shown by FOXL2 and CDKN1B expression in pre-granulosa cells throughout the gonad (a', a'', f', f'', yellow arrowheads). These fetal ovaries lacked mature granulosa cells expressing AMH (a). In contrast, XX *Rspo1^-/-^* (*Rspo1^KO^*), XX *Rspo1^-/-^; Sox8^-/-^* (*Rspo1^KO^ Sox8^KO^*), XX *Rspo1^-/-^; Sox9^flox/flox^; Sf1:cre^Tg/+^* (*Rspo1^KO^ Sox9^cKO^*), and XX *Rspo1^KO^ Sox8^KO^ Sox9^cKO^* gonads exhibited down-regulation of CDKN1B (b–e) and ectopic AMH expression in the anterior area (b'-e', asterisks), indicating Sertoli cells or mature granulosa cells. However, while XX *Rspo1^KO^* gonads contained Sertoli cells expressing DMRT1 (g', white arrowhead), these cells were rare in XX double and triple mutants (h–j) (for XX triple mutants, 1 out of 8 gonads studied from n = 4 fetuses). Note that some DMRT1-positive cells are germ cells expressing TRA98 (g'', h'', j'', plus symbols). Thus, while granulosa cells differentiate precociously in XX *Rspo1^KO^* gonads lacking *Sox8* and/or *Sox9* at E17.5, these cells have not yet reprogrammed as Sertoli cells in XX *Rspo1^KO^ Sox8^KO^* and XX *Rspo1^KO^ Sox9^cKO^* mice. In XX triple mutant fetuses, granulosa cell reprogramming as Sertoli cells may be delayed, or blocked.

The online version of this article includes the following figure supplement(s) for figure 3:

**Figure supplement 1.** Absence of SOX9 expression in gonads from XX *Rspo1^KO^ Sox8^KO^ Sox9^cKO^* fetuses and presence of steroidogenic cells at E17.5.

**Figure supplement 2.** Quantification of immunostained cells expressing DMRT1, FOXL2, and CDKN1B at E17.5.

indicating ovarian development. The granulosa cells remained quiescent, as evidenced by expression of the mitotic arrest marker CDKN1B (also known as P27) throughout the E17.5 gonad, and the absence of AMH expression indicated that these cells were fetal or pre-granulosa cells (*Figure 3a*; *Maatouk et al., 2013*).

In contrast, CDKN1B is down-regulated in the anterior area of XX *Rspo1^KO^ Sox8^KO^ Sox9^cKO^* triple mutant gonads (n = 4 triple mutant fetuses, *Figure 3e*), as in XX *Rspo1^KO^* single, as well as in XX *Rspo1^KO^ Sox8^KO^* and XX *Rspo1^KO^ Sox9^cKO^* double mutants (*Figure 3b,c,d*; *Maatouk et al., 2013*). In addition, these mutants contained cells expressing AMH (*Figure 3b'–e' asterisks*), indicating precocious granulosa cell differentiation, as previously described (*Maatouk et al., 2013*). However, while SOX9 and DMRT1 positive, TRA98-negative Sertoli cells were readily detectable in the anterior area of the XX *Rspo1^KO^* gonads (*Figure 3—figure supplement 1b'* and *Figure 3g'*, white

arrowheads), these cells were noticeably absent or rare in XX $Rspo1^{KO}$ $Sox8^{KO}$ $Sox9^{cKO}$ triple mutant fetuses (1 out of 8 XX triple mutant gonads studied from n = 4 fetuses) (*Figure 3—figure supplement 1e* and *Figure 3j*). This was also the case in XX $Rspo1^{KO}$ $Sox8^{KO}$ and XX $Rspo1^{KO}$ $Sox9^{cKO}$ double mutants (*Figure 3—figure supplement 1c,d* and *Figure 3h,i*). Together with these observations, quantification of immunostained cells expressing DMRT1, FOXL2, and CDKN1B per gonadal section area demarcated by DAPI (*Figure 3—figure supplement 2a–f*) confirmed the lack of Sertoli cells and presence of granulosa cells in XX double and triple mutant gonads at E17.5 (*Figure 3—figure supplement 2a,c,e*).

In addition to the presence of mature granulosa cells, gonads in the XX single, double, and triple mutant fetuses also exhibited NR5A1- and HSD3β-positive cells (*Figure 3—figure supplement 1g–j*), which were absent in XX control ovaries (*Figure 3—figure supplement 1f*; *Chassot et al., 2008*; *Lavery et al., 2012*). Thus, these data indicated that ablation of *Sox8* and/or *Sox9* in XX fetuses lacking *Rspo1* does not prevent the appearance of steroidogenic cells and precocious granulosa differentiation, two characteristics of XX $Rspo1^{KO}$ gonads (*Maatouk et al., 2013*; *Chassot et al., 2008*).

We then examined the phenotype of $Rspo1^{KO}$ $Sox8^{KO}$ $Sox9^{cKO}$ gonads in E17.5 XY fetuses (*Figure 4a–h'''* and *Figure 4—figure supplement 1a–h'*). As shown, XY $Rspo1^{KO}$ $Sox8^{KO}$ double mutant gonads contained SOX9 and DMRT1 positive Sertoli cells forming testis cords, as in control fetal testes (*Figure 4e,f* and *Figure 4—figure supplement 1a,b*). Also, XY $Rspo1^{KO}$ $Sox9^{cKO}$ gonads exhibited DMRT1-positive testis cords (*Figure 4g,g'*), which were more pronounced than testis cords in XX $Rspo1^{KO}$ and XX $Rspo1^{KO}$ $Sox9^{cKO}$ gonads at this stage (*Figure 3g,g',i,i'*; *Lavery et al., 2012*). Thus, in XY fetuses lacking *Rspo1*, inactivation of one *Sox* gene is dispensable for Sertoli cells. However, in fetuses lacking both *Sox8* and *Sox9* in XY triple mutant gonads, Sertoli cells expressing DMRT1 were not readily obvious (6 of 6 XY triple mutant gonads studied from n = 3 fetuses) (*Figure 4h* and *Figure 4—figure supplement 1d*). Instead, as in XY $Rspo1^{KO}$ $Sox9^{cKO}$ gonads at this stage, XY triple mutant gonads exhibited FOXL2-positive pre-granulosa cells (*Figure 4h,h',h'''*, yellow arrowheads), and AMH expression suggested that some mature granulosa cells were present (*Figure 4d'*, asterisk). Quantification of cells expressing DMRT1, FOXL2, and CDKN1B confirmed these observations (*Figure 3—figure supplement 2b,d,f*). Like XX triple mutants, XY triple mutants also contained steroidogenic cells expressing NR5A1 and HSD3β (*Figure 4—figure supplement 1h*).

Altogether, fetal XY and XX $Rspo1^{KO}$ $Sox8^{KO}$ $Sox9^{cKO}$ gonads resembled gonads from XX $Rspo1^{KO}$ $Sox8^{KO}$, as well as XY and XX $Rspo1^{KO}$ $Sox9^{cKO}$ fetuses, with respect to the presence of steroidogenic cells and mature granulosa cells. However, fetal triple mutant gonads lacked Sertoli cells that were present in fetal (*Figure 4f,g*) or post-natal (*Figure 1Bf,j, Cf,j*) double mutant mice. Thus, while pre-granulosa cells in triple mutants differentiated precociously, their reprogramming as Sertoli cells forming testis cords at E17.5 appears to be blocked, or delayed.

## Lack of Sertoli cell differentiation in XX and XY $Rspo1^{KO}$ $Sox8^{KO}$ $Sox9^{cKO}$ fetuses

In order to further address the development of triple mutant gonads, we extended our analyses to juvenile (P10) and adult (P40) mice (*Figure 5a–d'*; *Figure 5—figure supplement 1a–x*; *Figure 5—figure supplement 2a–j* and *Figure 5—figure supplement 3a–o'''*). Both XY and XX $Rspo1^{KO}$ $Sox8^{KO}$ $Sox9^{cKO}$ triple mutant mice developed externally as female with a short anogenital distance, as in XX control mice (*Figure 5c,p,r*). Internally, both XY and XX triple mutants displayed hermaphroditism of the reproductive tracts, as shown by concomitant presence of vasa deferensia and uteri (*Figure 5—figure supplement 1c,m*). This was also observed in XY and XX $Rspo1^{KO}$ $Sox9^{cKO}$ mice (*Figure 5—figure supplement 1b,n*), as well as in XX $Rspo1^{KO}$ and in XX $Rspo1^{KO}$ $Sox8^{KO}$ mice (*Figure 2—figure supplement 1i,j*). Histological analyses revealed that XY and XX triple mutant gonads developed as ovaries containing primary follicles at P10 (*Figure 5—figure supplement 1f,p*), which matured up to the antral follicle stage at P40, though some exhibited irregular granulosa cell organization (*Figure 5i,v*, *blue arrowheads*). The triple mutant gonads occasionally contained immature or atrophied follicles (*Figure 5—figure supplement 2a,b*). Both XY and XX $Rspo1^{KO}$ $Sox8^{KO}$ $Sox9^{cKO}$ gonads lacked testicular sex cords (*Figure 5i,v*, *Figure 5—figure supplement 1f,p*, and *Figure 5—figure supplement 2a,b*), which were found in XY and XX $Rspo1^{KO}$ mice lacking *Sox8* (*Figure 2f,q* and *Figure 2—figure supplement 1d,l*) or *Sox9* (*Figure 5h,w* and *Figure 5—figure supplement 1e,q*). Immunostaining experiments on P10 and P40 triple mutant gonads confirmed the presence of

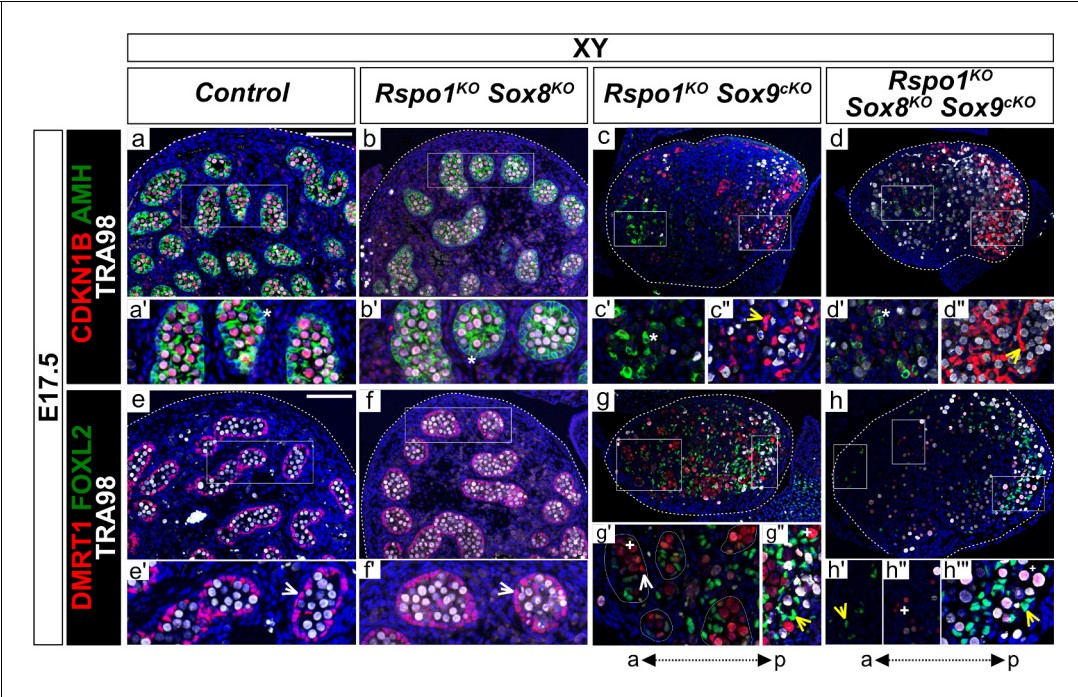

**Figure 4.** Lack of testis cords in XY *Rspo1^KO Sox8^KO Sox9^cKO* triple mutant fetuses at E17.5. Immunofluorescence of CDKN1B (P27) (mitotic arrest marker, in red) (a–d''), AMH (Sertoli marker and mature granulosa cell marker, in green) (a–d''), DMRT1 (Sertoli and germ cell marker, in red) (e–h''), FOXL2 (granulosa cell marker, in green) (e–h'''), TRA98 (germ cell marker, in white) (a–h'''), and DAPI (nuclear marker, in blue) (a–h) on gonadal sections from E17.5 fetuses (main panel scale bar 100 μm). For gonads in panels (c–d), the anterior 'a' and posterior 'p' axis is shown below each column. Below each main panels (a–h), highlighted areas are shown in respective primed letter panels. Yellow arrowheads indicate granulosa cells expressing CDKN1B or FOXL2, asterisks indicate cells expressing AMH, white arrowheads indicate Sertoli cells expressing DMRT1, and plus symbols indicate germ cells expressing DMRT1 and TRA98. Gonads in XY *Rspo1^-/-; Sox8^-/-* (*Rspo1^KO Sox8^KO*) fetuses exhibited AMH and DMRT1 positive Sertoli cells organized as testis cords (b, f) and lacked FOXL2-positive granulosa cells (f), as in control testes (a, e). Cells expressing AMH were found in XY *Rspo1^-/-; Sox9^flox/flox; Sf1:cre^Tg/+* (*Rspo1^KO Sox9^cKO*) and XY *Rspo1^KO Sox8^KO Sox9^cKO* gonads (c' and d', asterisks), indicating Sertoli cells or mature granulosa cells. Indeed, both exhibited granulosa cells expressing CDKN1B and FOXL2 (c', d', g', h', h''', yellow arrowheads). However, while XY *Rspo1^KO Sox9^cKO* gonads exhibited DRMT1-positive, TRA98-negative Sertoli cells (g', white arrowhead), these cells were scarce in XY triple mutant gonads (h) (6 of 6 XY triple mutant gonads studied from n = 3 fetuses). Note that some DMRT1 expressing cells in XY *Rspo1^KO Sox9^cKO* and XY triple mutant gonads are germ cells expressing TRA98 (g', g'', h'', h''', plus symbols). Thus, although XY *Rspo1^KO Sox9^cKO* and XY triple mutant gonads contain mature granulosa cells at E17.5, these cells do not reprogram as Sertoli cells in XY triple mutant fetuses.

The online version of this article includes the following figure supplement(s) for figure 4:

**Figure supplement 1.** Absence of SOX9 expression in gonads from XY *Rspo1^KO Sox8^KO Sox9^cKO* fetuses and presence of steroidogenic cells at E17.5.

follicles with granulosa cells expressing FOXL2 and the absence of testis cords with Sertoli cells expressing DMRT1 (*Figure 5l,o,y,b'* and *Figure 5—figure supplement 1i,l,s,v*).

In 3 of 10 XY and 6 of 16 XX post-natal gonads studied, a cluster of cells expressing DMRT1 were found, but further analyses revealed that these cells did not express the mature Sertoli cell marker GATA1 (*Beau et al., 2000*; *Figure 5—figure supplement 3f,f',f'',l,l',l'',o,o',o''*). Instead, these cells expressed the embryonic supporting cell marker GATA4 (*Tevosian et al., 2002*), which suggests rudimentary testis cord formation (*Figure 5—figure supplement 3c,c',c'',i,i',i''*, asterisks). We also noticed some cells expressing DMRT1 and FOXL2, though these cells were rare (*Figure 5—figure supplement 3l'',l'''*, arrowheads). In fact, immunostaining for FOXL2 confirmed that the vast majority of the supporting cells in triple mutants were granulosa cells, which did not undergone reprogramming into Sertoli cells (*Figure 5l,o,y,b'* and *Figure 5—figure supplement 1i,l,s,v*).

While observing atrophied follicles in adult XY and XX *Rspo1^KO Sox8^KO Sox9^cKO* triple mutant mice, a distinct interstitial compartment was also apparent (*Figure 5i,v*, asterisks and *Figure 5—figure supplement 2a,b*). The identity of this compartment was confirmed by immunostaining for NR5A1 and HSD3β (*Figure 5—figure supplement 2g,h*). In triple mutant gonads, the interstitial cells were arranged individually or in small clusters when compared with XX control ovaries and XX

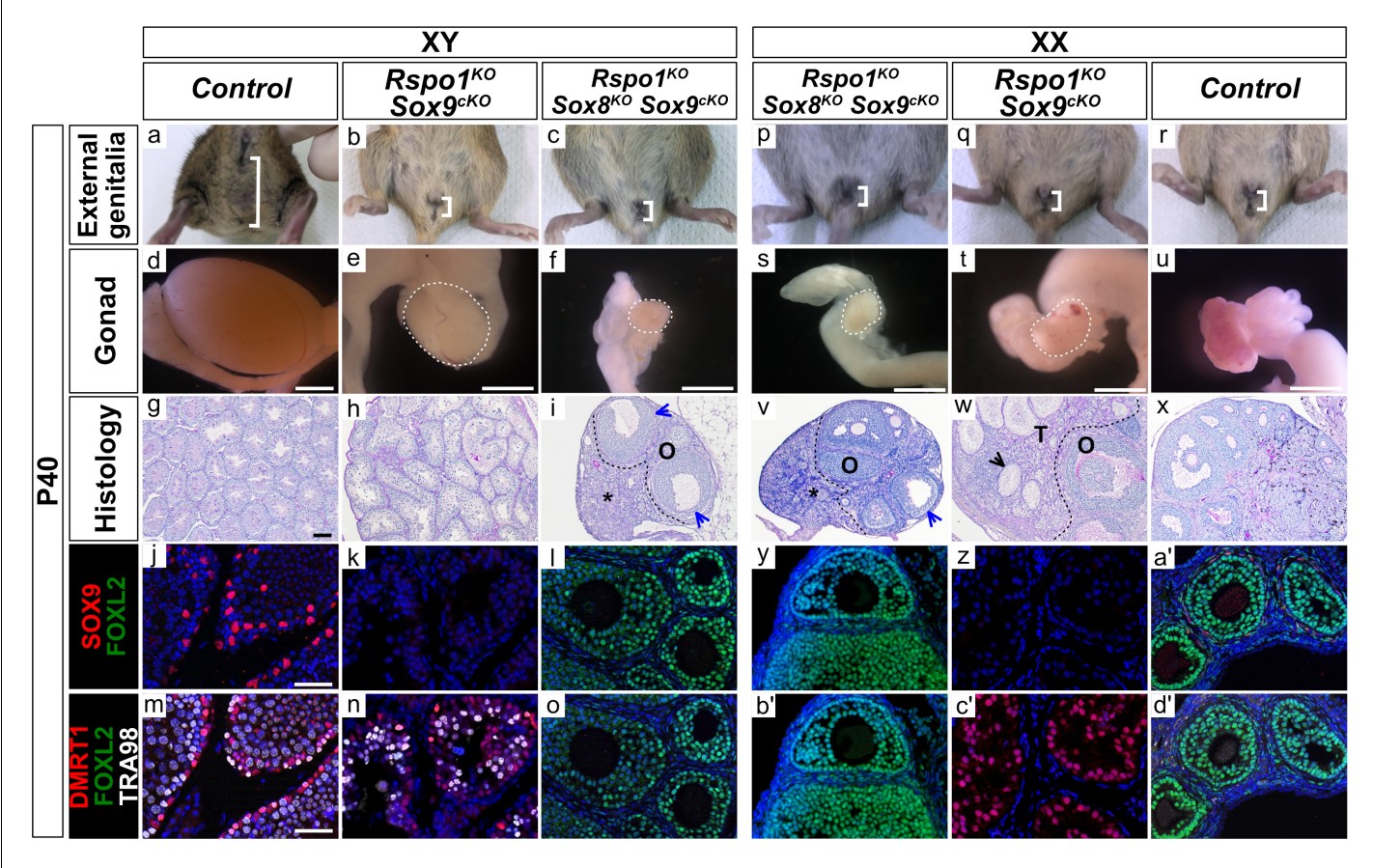

**Figure 5.** Absence of seminiferous tubules in XY and XX *Rspo1^KO^ Sox8^KO^ Sox9^cKO^* triple mutant adult mice. External genitalia from adult P40 mice (a–c, p–r), macroscopic view of gonads (d–f, s–u) (Scale bars 1.5 mm), histology as revealed by PAS staining on gonadal sections (g–i, v–x) (Scale bars 100 µm), and immunostaining of SOX9 (Sertoli cell marker, in red) (j–l, y–a'), FOXL2 (granulosa cell marker, in green) (j–o, y–d'), DMRT1 (Sertoli and germ cell marker, in red) (m–o, b'–d'), TRA98 (germ cell marker, in white) (m–o, b'–d'), and DAPI (nuclear marker, in blue) (j–o, y–d') on gonadal sections (Scale bars 50 µm). As shown, though adult XY *Rspo1^-/-^; Sox9^fl/fl^; Sf1:cre^Tg/+^* (*Rspo1^KO^ Sox9^cKO^*) double mutant gonads developed a short anogenital distance (b), internally these mice developed hypoplastic testes (compare e with d). XX *Rspo1^KO^ Sox9^cKO^* gonads developed as ovotestes (t), as in XX *Rspo1^KO^* single mutants and XX *Rspo1^KO^ Sox8^KO^* double mutants (see **Figure 2r, q**). Although *Sox9^fl/fl^* is inactivated by *Sf1:cre^Tg/+^* in *Rspo1^KO^ Sox9^cKO^* mice (k, z), XY double mutant gonads exhibited seminiferous tubules (h) containing DMRT1-positive Sertoli cells which are TRA98-negative (n), as in control testes (g, m). XX double mutant gonads contained an ovarian compartment 'O' with follicles and a testicular 'T' compartment with seminiferous tubule-like structures, as indicated by black arrowheads (w). The seminiferous tubule-like structures contained DMRT1-positive Sertoli cells (c'), as in control testes (m), but lacked TRA98-positive germ cells (c'). Both the XY and XX *Rspo1^KO^ Sox8^KO^ Sox9^cKO^* triple mutant mice develop externally as female with a short anogenital distance (c, p) as in the double mutants and control female (b, q–r). Despite this, the triple mutants gonads (f, s) developed as atrophied ovaries (i, v), which were smaller than control ovaries (u, x). XY and XX triple mutant gonads exhibited an ovarian 'O' compartment and a distinct interstitial compartment, as indicated by asterisks (i, v). The gonads contained follicles up to the antral stage, though some exhibited irregular granulosa cell organization, as indicated by blue arrowheads (i, v). Notably, XY and XX triple mutants lacked testicular sex cords (i, v) that were present in XY and XX *Rspo1^KO^ Sox9^cKO^* gonads (h, w). Immunostaining on XY and XX *Rspo1^KO^ Sox8^KO^ Sox9^cKO^* gonads confirmed the absence of SOX9 and DMRT1 positive Sertoli cells and the presence of ovarian follicles with granulosa cells expressing FOXL2 (l, o, y, b'), as in control ovaries (a', d'). For these analyses, n = 3 XY and n = 3 XX triple mutant mice were examined.

The online version of this article includes the following figure supplement(s) for figure 5:

**Figure supplement 1.** Reproductive tract of XY and XX *Rspo1^KO^ Sox8^KO^ Sox9^cKO^* adult P40 mice and analyses in juvenile P10 mice.

**Figure supplement 2.** Organization of adult XY and XX *Rspo1^KO^ Sox8^KO^ Sox9^cKO^* triple mutant gonads.

**Figure supplement 3.** Rare immature Sertoli cells in XY and XX *Rspo1^KO^ Sox8^KO^ Sox9^cKO^* triple mutant gonads.

*Rspo1^KO^* ovotestes. In addition, XY and XX triple mutant interstitial cells mildly atrophied, appeared collapsed/dysplastic, and lacked interstitial sinusoids (*Figure 5—figure supplement 2e,f*). No evidence of neoplasia was present in XY and XX triple mutant and in XX *Rspo1^KO^* gonads.

In summary, gonads in XY and XX triple mutants developed as atrophied ovaries. Altogether, our data clearly demonstrate that *Sox8* or *Sox9* is required and sufficient for testicular differentiation in XY and XX *Rspo1^{KO} Sox9^{cKO}* or *Rspo1^{KO} Sox8^{KO}* double mutants, respectively.

## Discussion

Our results emphasize the essential role of SOX genes in testis differentiation as we show that *Sox* genes are required for Sertoli cell differentiation in XX ovotestis. The critical domain of SOX proteins is the DNA binding domain, the HMG (High-Mobility Group)-domain that binds in a sequence-specific manner (*Mertin et al., 1999*). Remarkably, an HMG-box gene is associated with male sex-specific region in the brown algae *Ectocarpus* (*Ahmed et al., 2014*). The sexual cycle of this species consists of an alternation between a diploid sporophyte (with both the U and the V chromosomes), which after meiosis produces either a female haploid gametophyte (with the U chromosome) or male gametophyte (with the V chromosome). The sex-specific region of the *Ectocarpus* V-chromosome contains an HMG-domain gene, suggesting a conserved function of the HMG-domain containing genes in maleness throughout evolution. In mice, when the HMG box of SRY is replaced with that of SOX3 or SOX9, these composite *Sox* transgenes induce *Sox9* expression and Sertoli cell differentiation (*Bergstrom et al., 2000*). Also, transgenic expression of *Sox3* or *Sox10* in XX gonads results in *Sox9* expression and testicular differentiation (*Sutton et al., 2011*; *Polanco et al., 2010*). These examples demonstrated functional conservation among *Sox* genes or HMG-box domains and also suggests that male fate centers on transactivation of *Sox9*.

However, testicular differentiation was reported in XY and XX *Rspo1/Ctnnb1 Sox9* double mutant mice (*Nicol and Yao, 2015*; *Lavery et al., 2012*), suggesting that another *Sox* gene can substitute for the absence of *Sox9* in this context. Given that *Sox8* is up-regulated in the double mutant gonads (*Nicol and Yao, 2015*; *Lavery et al., 2012*), we hypothesized that *Sox8* and *Sox9* can act redundantly for testicular development in mice lacking *Rspo1*. Here, we demonstrated this by showing that in XY and XX *Rspo1^{KO}* mice: (*i*) *Sox8* and *Sox9* are expressed independently; (*ii*) *Sox8* or *Sox9* is sufficient for Sertoli cell differentiation in *Rspo1^{KO} Sox9^{cKO}* and *Rspo1^{KO} Sox8^{KO}* mice, respectively; and (*iii*) *Sox8* or *Sox9* are required for testicular differentiation, as evidenced by the development of atrophied ovaries in *Rspo1^{KO} Sox8^{KO} Sox9^{cKO}* triple mutant mice. Together our data show that *Sox8* is able to substitute for *Sox9* to induce Sertoli cell differentiation in XX sex reversal.

The gonad fate in wildtype, *Sox* and *Rspo1* mutant mice is summarized in *Figure 6*. In wildtype mice, SOX9 promotes testicular differentiation in XY gonads and RSPO1 promotes ovarian differentiation in XX gonads (*Figure 6a*). This is also the case in mice lacking *Sox8*, since it is dispensable for testis and ovarian development (*Figure 6a*; *Sock et al., 2001*). As shown, there is an antagonistic relationship between the testis and ovarian pathways, such that the activation of one pathway also leads to the repression of the other to ensure one gonadal fate (*Figure 6a*). In XY *Sox9^{cKO}* mice, the testis pathway is not activated, and the ovarian pathway is not repressed, leading to ovarian differentiation (*Figure 6b*). In XX *Sox9^{cKO}* mice, loss of SOX9 does not impair ovarian development (*Figure 6b*). In XY *Rspo1^{KO} Sox8^{KO}* or XY *Rspo1^{KO} Sox9^{cKO}* mice, gonads develop as testes or hypoplastic testes, since one SOX factor is sufficient for Sertoli cell differentiation and seminiferous tubule formation (*Figure 6c,d*). This is also exemplified by ovo-testicular development in XX *Rspo1^{KO} Sox8^{KO}* and XX *Rspo1^{KO} Sox9^{cKO}* mice (*Figure 6c,d*), where Sertoli cells arise from reprogramming of pre-granulosa cells that have precociously differentiated (*Maatouk et al., 2013*). We found that inactivation of both SOX factors in mice lacking RSPO1 prevents testicular development in XY and XX animals. In XY and XX *Rspo1^{KO} Sox8^{KO} Sox9^{cKO}* triple mutant embryos, though pre-granulosa cells differentiate precociously, the absence of both SOX factors impedes granulosa-to-Sertoli reprogramming in embryos and gonads develop as atrophied ovaries (*Figure 6e*). This atrophied ovary outcome suggests that FOXL2 and other ovarian factors cannot fully compensate for the loss of RSPO1 (*Figure 6e*).

Interestingly, in XX *Rspo1^{KO}* single mutant gonads and in XX *Rspo1^{KO}* gonads lacking *Sox8* and/or *Sox9* in double and triple mutants, pre-granulosa cells differentiate as mature granulosa cells expressing AMH in an anterior-to-posterior wave or gradient ([*Maatouk et al., 2013*] and present data). Such a gradient was also found in XX gonads with an *Sry* transgene – supporting cells transiently express SOX9, after which this ability is lost in an anterior-to-posterior wave (*Harikae et al.,*

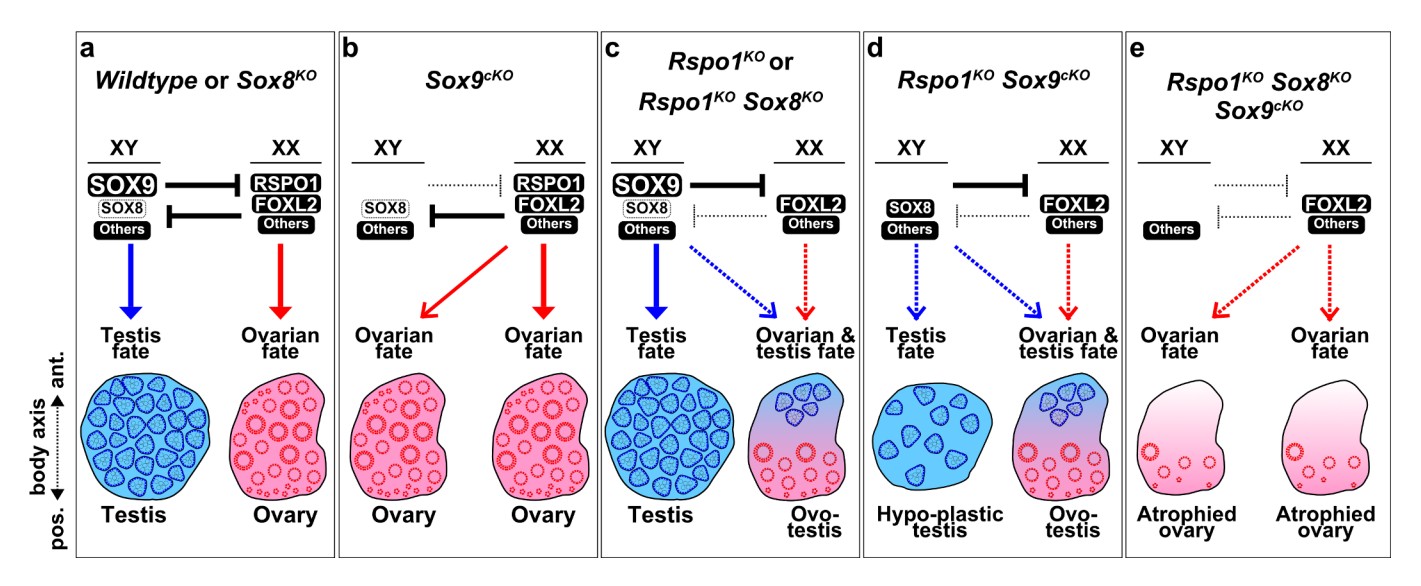

**Figure 6.** Gonad fate in wildtype, *Rspo1*, and *Sox* mutant mice. In wildtype mice, SOX9, SOX8, and other factors promote testicular differentiation in XY mice, and RSPO1, FOXL2, and other factors promote ovarian differentiation in XX mice, as indicated by arrows (**a**). Antagonism exists between the testis and ovarian pathway, as indicated by 'T' bars (**a**). SOX9 and RSPO1 are essential for testicular and ovarian differentiation respectively, since XY *Sox9^cKO* mice develop ovaries (**b**) and XX *Rspo1^KO* mice develop partial sex-reversal ovotestes (**c**). However, we previously demonstrated that SOX9 is dispensable for testicular development in XX *Rspo1^KO* mice, by studying XX *Rspo1^KO Sox9^cKO* mice (**d**). Also, gonads in XY *Rspo1^KO Sox9^cKO* mice develop as hypoplastic testes (**d**), indicating that RSPO1 is required for ovarian differentiation in XY *Sox9^cKO* mice. In studying a *Sox8^KO* mutation in XY and XX mice or mice lacking *Rspo1*, it was evident that SOX8 is dispensable for testicular, ovarian, or ovotesticular development (**a, c**). In this study however, we demonstrated that SOX8 is required for hypoplastic testicular or ovotesticular differentiation in XY and XX *Rspo1^KO Sox9^cKO* mice (**d**) by studying triple mutants (**e**). Gonads in both XY and XX *Rspo1^KO Sox8^KO Sox9^cKO* mice lacked testis cords and developed as atrophied ovaries (**e**). Thus, SOX8 or SOX9 is sufficient and both SOX are required for testicular differentiation in gonads lacking RSPO1.

*2013*). This suggests that somatic cell differentiation in ovaries proceeds in a spatiotemporal, anterior-to-posterior, manner. As shown here, apparently, this wave of somatic cell differentiation is conserved in XX sex reversal associated with *Rspo1* mutations.

How *Sox8* operates in pathophysiological cases of testicular differentiation is not yet known. In wildtype mice, *Sox8* expression in XY gonads has been described as coinciding with *Sox9* at E11.5 (*Jameson et al., 2012*; *Stévant et al., 2018*; *Schepers et al., 2003*) or occurring after robust expression of *Sox9* at E12.5 (*Schepers et al., 2003*). Together, these observations suggest that SRY might activate *Sox8*, as predicted (*Li et al., 2014*), and that *Sox8* expression is reinforced by *Sox9*. However, the activation of *Sox8* by SOX9 is likely indirect given that SOX9 does not bind the *Sox8* locus in mice. Interestingly, SOX9 binding to *Sox8* has been shown in cattle (*Rahmoun et al., 2017*). Also, the expression of *Sox8* and *Sox9* are independent in sex cords in XY mouse gonads ((*Barrionuevo et al., 2009*), our present results). Thus, it is more plausible that SRY activates *Sox8* expression in XY *Rspo1/Ctnnb1 Sox9* double mutant mice. In fact, in these mice, *Sry* expression is extended beyond E12.5 (*Lavery et al., 2012*; *Nicol and Yao, 2015*), a time when *Sry* is normally down-regulated in mice (*Hacker et al., 1995*).

Whereas *Sox8* expression can result from SRY activation in XY embryos, it is not obvious how *Sox8* is upregulated in the absence of *Sry* in XX *Rspo1/Ctnnb1 Sox9* double mutants. The pro-ovarian factors *Rspo1* or *Ctnnb1* are required on one hand, to prevent precocious maturation of granulosa cells, which are capable of transdifferentiation into Sertoli cells. On the other hand, these factors repress ectopic steroidogenesis in XX gonads as evidenced by the presence of steroidogenic cells in XX *Rspo1* or *Rspo1^KO Sox9^cKO* embryonic gonads (*Chassot et al., 2008*; *Lavery et al., 2012*).

When E13.5 ovaries are transplanted to kidneys of XY mice, circulating androgens promote partial trans-differentiation towards a testis fate through a mechanism involving up-regulation of *Sox8* before *Sox9* (*Miura et al., 2019*). Ablation of *Sox8* in the transplanted ovaries did not prevent sex

reversal and this outcome is likely attributed to the presence of *Sox9*. Furthermore, before up-regulation of *Sox8*, supporting cells in the fetal ovary transplant express *Amh*, a phenotype that is strikingly similar to sex reversal in XX *Rspo1KO* gonads (*Maatouk et al., 2013*). However, inactivation of *Amh* in the transplanted ovaries was also dispensable for sex reversal, suggesting that TGF-β signaling driven by other TGF-β factors like Activin or unknown factors may promote *Sox* gene expression in sex reversal conditions.

It is noteworthy that WNT/CTNNB1 signaling regulates the level of ActivinB as evidenced by its up-regulation in XX *Wnt4KO* or *Ctnnb1cKO* gonads (*Yao et al., 2006*; *Liu et al., 2010*). Moreover ablation of InhibinA/B, two antagonist members of Activin, promotes sex cord development in XX gonads (*Matzuk et al., 1992*). Thus, TGF-β signaling is likely involved in XX sex reversal when the WNT/CTNNB1 pathway is compromised. The factors including Activin/Inhibin to control *Sox* gene expression in XX transplanted ovaries and in XX mice lacking *Rspo1/Ctnnb1* remain to be identified.

The identification of *Sox8* as a key factor in pathophysiological testicular development is somewhat of a paradox, given evidence indicating that aside from *Sry* and *Sox9*, no other *Sox* gene tested so far play key roles in Sertoli cell differentiation in XY wildtype gonads (*She and Yang, 2017*). In mice, *Sox8* is dispensable for Sertoli cell differentiation (*Sock et al., 2001*), but an inefficient or late deletion of *Sox9* leads to XY sex reversal only if there is additional deletion of *Sox8* (*Lavery et al., 2011*; *Chaboissier et al., 2004*; *Barrionuevo et al., 2009*). This suggests that *Sox8* reinforces *Sox9* during testis differentiation. In addition, *Sox8* is required for Sertoli cell maintenance along with *Sox9*, since Sertoli cells in XY *Sox8 Sox9* double loss-of-function gonads undergo apoptosis (*Barrionuevo et al., 2016*). Thus, although *Sox8* is an important factor for testis differentiation and maintenance, the rapid and high level of expression of *Sox9* induced by SRY, minimizes the role of *Sox8* in XY differentiating testes.

During XX sex reversal, early and high induction of *Sox8/9* does not occur, given the absence of *Sry*. Hence, ovarian differentiation is initiated, but absence of pro-ovarian gene such as *Rspo1* leads to a succession of events from ectopic steroidogenesis to accelerated maturation of granulosa cells that ultimately promote the expression of *Sox8* and *Sox9*. In XX *Rspo1KO* gonads, both factors are similarly important and can compensate for the absence of the other to induce transdifferentiation of granulosa cells to Sertoli cells during late embryogenesis. Thus, although *Sox8* is dispensable for testicular differentiation in wildtype mice, our current study demonstrates that *Sox8* is essential for testicular differentiation in sex reversal conditions.

Functional redundancy between SOX8 and SOX9 does not seem to operate in humans. For example, XY sex reversal can result from inactivating mutations of one *SOX9* allele, indicating haploinsufficiency (*Wagner et al., 1994*; *Foster et al., 1994*). Also, *SOX8* mutations were associated with a range of phenotypes including complete gonadal dysgenesis (streak gonads with immature female genitalia) and hypoplastic testes in three 46, XY patients (*Portnoi et al., 2018*). Thus, it appears that the impact of a single gene mutation can vary, according to the nature of the mutation and genetic background of the individual. Nevertheless, the human cases of XY sex reversal show that SOX8 is emerging to be an important regulator of testicular gonadal development and by extension, overall male development.

## Materials and methods

### Key resources table

| Reagent type (species) or resource | Designation | Source or reference | Identifiers | Additional information |
|---|---|---|---|---|
| Strain, strain (*Mus musculus*) | *Rspo1-/-* | (*Chassot et al., 2008*) | | henceforth *Rspo1KO* |
| Strain, strain (*Mus musculus*) | *Sox8-/-* | (*Sock et al., 2001*) | | henceforth *Sox8KO* |
| Strain, strain (*Mus musculus*) | *Sox9fl/fl; Sf1:creTg/+* | (*Lavery et al., 2011*) | | henceforth *Sox9cKO* |
| Strain, strain (*Mus musculus*) | *Rspo1-/-; Sox8-/-* | This paper | | henceforth *Rspo1KO Sox8KO* |

*Continued on next page*

*Continued*

| Reagent type (species) or resource | Designation | Source or reference | Identifiers | Additional information |
|---|---|---|---|---|
| Strain, strain (*Mus musculus*) | *Rspo1$^{-/-}$; Sox9$^{fl/fl}$; Sf1:cre$^{Tg/+}$* | (*Lavery et al., 2012*) | | henceforth *Rspo1$^{KO}$ Sox9$^{cKO}$* |
| Strain, strain (*Mus musculus*) | *Rspo1$^{-/-}$; Sox8$^{-/-}$; Sox9$^{fl/fl}$; Sf1:cre$^{Tg/+}$* | This paper | | henceforth *Rspo1$^{KO}$ Sox8$^{KO}$ Sox9$^{cKO}$* |
| Antibody | anti-AMH/MIS (C-20) (Goat polyclonal) | Santa Cruz RRID:AB-649207 | Cat# sc-6886 | IF(1:100) |
| Antibody | anti-DMRT1 (Rabbit polyclonal) | Sigma RRID:AB_10600868 | Cat# HPA027850 | IF(1:100) |
| Antibody | anti-FOXL2 (Goat polyclonal) | Novus RRID:AB_2106188 | Cat# NB100-1277 | IF(1:200) |
| Antibody | anti-GATA1 (N6) (Rat monoclonal) | Santa Cruz RRID:AB_627663 | Cat# sc-265 | IF(1:200) |
| Antibody | anti-GATA4 (C20) (Goat polyclonal) | Santa Cruz RRID:AB_2108747 | Cat# sc-1237 | IF(1:200) |
| Antibody | anti-3β-HSD (P18) (Goat polyclonal) | Santa Cruz RRID:AB_2279878 | Cat# sc-30820 | IF(1:200) |
| Antibody | anti-P27/CDKN1B (Kip1) (Rabbit polyclonal) | Santa Cruz RRID:AB_632129 | Cat# sc-528 | IF(1:200) |
| Antibody | anti-Laminin LAMA1 (Rabbit polyclonal) | Sigma RRID:AB_477163 | Cat# L9393 | IF(1:150) |
| Antibody | anti-NR5A1/SF-1 (Rabbit polyclonal) | Gift from Ken Morohashi | | IF(1:1000) |
| Antibody | anti-SOX8 (Guineapig, polyclonal) | Gift from Elisabeth Sock (*Stolt et al., 2005*) | | IF(1:1000) |
| Antibody | anti-SOX9 (Rabbit polyclonal) | Sigma RRID:AB_1080067 | Cat# HPA001758 | IF(1:200) |
| Antibody | anti-TRA98 (Rat monoclonal) | Abcam RRID:AB_1659152 | Cat# ab82527 | IF(1:200) |
| Recombinant DNA reagent | *Rspo1* riboprobe | (*Parma et al., 2006*) | | |
| Recombinant RNA reagent | *Rspo1* (RNAscope riboprobe) | Advanced Cell Diagnostics | | |
| Software, algorithm | Affinity Photo | Serif Europe Ltd., Nottingham United Kingdom | | https://affinity.serif.com/en-us/photo/ |
| Software, algorithm | Affinity Designer | Serif Europe Ltd., Nottingham United Kingdom | | https://affinity.serif.com/en-us/ |
| Software, algorithm | Graphpad Prism | Graphpad Software, Inc, La Jolla, CA | | http://www.graphpad.com/ |
| Other | DAPI stain | Vector Laboratory | H-1500 | (1µg/ml) |

## Mouse strains and genotyping

The experiments described here were carried out in compliance with the relevant institutional and French animal welfare laws, guidelines, and policies. These procedures were approved by the French ethics committee (Comité Institutionnel d'Ethique Pour l'Animal de Laboratoire; number NCE/2011–12). All mouse lines were kept on a mixed 129Sv/C57BL6/J background. *Rspo1$^{-/-}$* (*Chassot et al., 2008*), *Sox8$^{-/-}$* (*Sock et al., 2001*), *Sox9$^{fl/fl}$* (*Akiyama et al., 2002*), and *Sf1:cre$^{Tg/+}$* (*Bingham et al., 2006*) mice were obtained previously, and the generation of *Sox9$^{fl/fl}$; Sf1:cre$^{Tg/+}$* (*Lavery et al., 2011*) and *Rspo1$^{-/-}$; Sox9$^{fl/fl}$; Sf1:cre$^{Tg/+}$* (*Lavery et al., 2012*) mice was described previously. For *Rspo1$^{KO}$ Sox8$^{KO}$* mice: *Rspo1$^{-/-}$* males were mated with *Sox8$^{-/-}$* females to obtain *Rspo1$^{+/-}$; Sox8$^{+/-}$* males and females. Matings between these littermates allowed us to obtain *Rspo1$^{-/-}$; Sox8$^{-/-}$* double mutant mice, referred to as *Rspo1$^{KO}$ Sox8$^{KO}$* mice, and control animals. For *Rspo1$^{KO}$ Sox8$^{KO}$*

$Sox9^{cKO}$ mice: first, $Rspo1^{-/-}$; $Sox8^{-/-}$ males were mated with $Sox8^{-/-}$; $Sox9^{fl/fl}$; $Sf1:cre^{Tg/+}$ females to generate $Rspo1^{+/-}$; $Sox8^{-/-}$; $Sox9^{fl/+}$ males and $Rspo1^{+/-}$; $Sox8^{-/-}$; $Sox9^{fl/+}$; $Sf1:cre^{Tg/+}$ females. Matings between these littermates then produced $Rspo1^{-/-}$; $Sox8^{-/-}$; $Sox9^{fl/fl}$ males and $Rspo1^{+/-}$; $Sox8^{-/-}$; $Sox9^{fl/fl}$; $Sf1:cre^{Tg/+}$ females. Finally, matings between these littermates then allowed us to obtain $Rspo1^{-/-}$; $Sox8^{-/-}$; $Sox9^{fl/fl}$; $Sf1:cre^{Tg/+}$ triple mutant mice, referred to as $Rspo1^{KO}$ $Sox8^{KO}$ $Sox9^{cKO}$ mice, and control animals. Embryos were collected from timed evening matings that was confirmed by the presence of a vaginal plug the following morning. This marked embryonic day 0.5 (E0.5). The day of delivery was defined as post-natal day 0 (P0). Genotyping was performed as described in *Chaboissier et al. (2004)*; *Chassot et al. (2008)*; *Bingham et al. (2006)* by using DNA extracted from tail tip or ear biopsies of mice. The presence of the Y chromosome was determined, as described previously (*Hogan et al., 1994*).

## In situ hybridization

Gonad samples were fixed with 4% paraformaldehyde overnight, processed for paraffin embedding, and then sectioned at 5–7 µm thick. The in situ hybridizations for *Figure 1e–h* were carried out essentially as described by *Lavery et al. (2012)*. For analyses in *Figure 1a–d*, RNAscope technology was used (*Wang et al., 2012*). The *Rspo1* probe was purchased from the manufacturer (Advanced Cell Diagnostics) and the protocol was performed according to the manufacturer's instructions using the Fast Red dye, which can be visualized using light or fluorescence microscopy. The in situ hybridization experiments were performed on gonads from at least three mice for each genotype.

## Immunological analyses

Gonad samples were fixed with 4% paraformaldehyde overnight, processed for paraffin embedding, and sectioned at 5 µm thick. The following dilutions of primary antibodies were used: AMH/MIS (c-20, sc-6886, Santa Cruz), 1:200; DMRT1 (HPA027850, Sigma), 1:100; FOXL2 (NB100-1277, Novus), 1:200; GATA1 (N6, sc-265, Santa Cruz), 1:200; GATA4 (C20, sc-1237, Santa Cruz), 1:200; 3βHSD (P18, sc-30820, Santa Cruz), 1:200; P27 (Kip1, sc-528, Santa Cruz), 1:200; LAMA1 (L9393, Sigma), 1:150; SF1 (kindly provided by Ken Morohashi), 1:1000; SOX8 (kindly provided by Elisabeth Sock [*Stolt et al., 2005*]), 1:1000; SOX9 (HPA001758, Sigma), 1:200; and TRA98 (ab82527, Abcam), 1:200. Counterstain with DAPI was used to detect nuclei. Immunofluorescence of secondary antibodies were detected with an Axio ImagerZ1 microscope (Zeiss) coupled to an Axiocam mrm camera (Zeiss) or a LSM 780 NLO inverted Axio Observer.Z1 confocal microscope (Carl Zeiss Microscopy GmbH, Jena,Germany) using a Plan Apo 10X dry NA 0.45 objective. Images were processed with Axiovision LE and Serif Affinity Photo software. Immunostaining experiments were performed on gonads from at least three mice for each genotype.

## Cell quantification

Immunostaining analyses were performed, as described above. For analyses at E17.5, immunostaining were performed on 2 to 17 sections spaced 20–30 µm apart in each gonad. Then, for each section, the ratio of cells positive for DMRT1, FOXL2, or CDKN1B to total gonad area, as visualized by DAPI staining, were manually tabulated. Next, the individual ratios and mean for each genotype were plotted in a histogram using Graphpad software. Finally, the data was analysed by one-way ANOVA and Tukey-Kramer post tests. For p-values<0.05,<0.01,<0.001, and <0.0001, asterisks (*, **, ***, ****) represent significant differences compared with XY control cell numbers, respectively and ampersands (&, &&, &&&, &&&&) represent significant differences compared with XX control cell numbers, respectively.

## Histological analyses

Gonad samples were fixed with Bouin's solution overnight, processed for paraffin embedding, sectioned at 5 µm thick, and then stained according to standard procedures for periodic acid Schiff (PAS) or hematoxylin and eosin (H and E) staining. Images were taken with an Axiocam mrm camera (Zeiss) and processed with Serif Affinity Photo software. Histology staining was performed on gonads from at least three mice for each genotype.

## Acknowledgements

We thank Ken Morohashi and Elisabeth Sock for the SF-1/NR5A1 and SOX8 antibodies, respectively. We also thank Samah Rekima and the Experimental Histopathology Platform for assistance with samples. The microscopy was done at MICA facility and the help of Simon Lachambre is acknowledged. Thanks to the Schedl team for helpful discussions, and Anne-Amandine Chassot and Aitana Perea-Gomez for critical reading of the manuscript. This work was supported by the Agence Nationale de la Recherche (ANR-11-LABX-0028–01; ANR-19-CE14-0022-*SexDiff*).

## Additional information

### Funding

| Funder | Grant reference number | Author |
| --- | --- | --- |
| Agence Nationale de la Recherche | ANR-11-LABX-0028-01 | Nainoa Richardson |
| Agence Nationale de la Recherche | Graduate student fellowship | Nainoa Richardson |
| Agence Nationale de la Recherche | ANR-19-CE14-0022-SexDiff | Marie-Christine Chaboissier |

The funders had no role in study design, data collection and interpretation, or the decision to submit the work for publication.

### Author contributions

Nainoa Richardson, Conceptualization, Data curation, Formal analysis, Investigation, Methodology, Writing - original draft; Isabelle Gillot, Formal analysis, Investigation, Methodology, Writing - review and editing; Elodie P Gregoire, Investigation, Methodology; Sameh A Youssef, Dirk de Rooij, Alain de Bruin, Formal analysis; Marie-Cécile De Cian, Conceptualization, Data curation, Formal analysis, Supervision, Validation, Writing - review and editing; Marie-Christine Chaboissier, Conceptualization, Data curation, Formal analysis, Supervision, Funding acquisition, Validation, Project administration, Writing - review and editing

### Author ORCIDs

Nainoa Richardson https://orcid.org/0000-0003-1439-9764
Elodie P Gregoire https://orcid.org/0000-0001-7614-4754
Dirk de Rooij http://orcid.org/0000-0003-3932-4419
Marie-Christine Chaboissier https://orcid.org/0000-0003-0934-8217

### Ethics

Animal experimentation: This study was performed in strict accordance with the relevant institutional and European animal welfare laws, guidelines, and policies. These procedures were approved by the French ethics committee (Comité Institutionnel d'Ethique Pour l'Animal de Laboratoire; number NCE/2011-12).

### Decision letter and Author response

Decision letter https://doi.org/10.7554/eLife.53972.sa1
Author response https://doi.org/10.7554/eLife.53972.sa2

## Additional files

### Supplementary files

• Transparent reporting form

## Data availability

All data generated or analysed during this study are included in the manuscript and supporting files.

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
