## [Decision Letter]

**Acceptance summary:**

Your use of mice triple mutant for *Rspo1, Sox8* and *Sox9* clearly shows that in the absence of the female pathway driver, *Rspo1*, XX gonads form testes if either *Sox8* or *Sox9* is present, but not if both are absent. Even though *Sox8* does not appear to be a critical male determinant in the testis, during female to male sex reversal, *Sox8* is a more important player than was previously appreciated, more similar to the importance of *Sox8* in human sex determination.

**Decision letter after peer review:**

Thank you for submitting your article "*Sox8* and *Sox9* act redundantly for ovarian-to-testicular fate reprogramming in the absence of *Rspo1* in mouse sex reversal" for consideration by *eLife*. Your article has been reviewed by three peer reviewers, one of whom is a member of our Board of Reviewing Editors, and the evaluation has been overseen by Kathryn Cheah as the Senior Editor. The reviewers have opted to remain anonymous.

The reviewers have discussed the reviews with one another and the Reviewing Editor has drafted this decision to help you prepare a revised submission.

Summary:

Chaboissier showed that *Sox8* is not required during testis determination in mice (2004). Loss of *Sox9* alone results in ovarian development in XY mice, whereas loss of *Sox8* results in some defects in testis development but not sex reversal. However, in humans, mutations in SOX8 are associated with gonadal dysgenesis or hypoplastic testes, despite normal SOX9 function. Therefore, the importance of *Sox8* in the testis pathway has been in question.

This work follows on previous analyses by this lab and the Yao lab, showing that loss of *Rspo1* or *Ctnnb1* in the context of loss of *Sox9* still resulted in testis development in XX animals, suggesting that another gene was responsible for driving testis development. In this manuscript, the authors use triple mutants to show that in the absence of the female pathway driver, *Rspo1*, XX gonads form testes if either *Sox8* or *Sox9* is present, but not if both are absent. These results show that, at least in these circumstances, *Sox8* is a more important player than was previously appreciated.

These studies are technically impressive: Double knockouts are relatively common in the field, as a way of probing the relationship between pro-testis and pro-ovary networks and the consequence of missing key factors in both pathways. Triple KOs are much rarer. Such studies are difficult, mainly due to the rarity of fetuses carrying all of the appropriate alleles.

The data are clear and support the major claim, which is that SOX8 and SOX9 act redundantly to promote testicular tissue formation by sex reprogramming of precociously differentiated granulosa cells in XX *Rspo1^KO^* gonads. Along the way, there are a wealth of useful data reported of different combinations of mutant gonads, associated phenotypes and the expression of key gene/proteins in these.

Essential revisions:

1) This analysis was an enormous amount of work, and the results are surprising: Since *Sox8* is not critical to establish the testis pathway in the XY gonad, it is interesting that it is important to shift the XX gonad to the testis pathway. The authors primarily describe the change in expression of markers of the male and female pathways, and the physiological consequences on secondary sex characteristics, but do not get at why *Sox8* is more important to drive testis fate in the XX gonad than it is to drive testis fate in the XY gonad (in mice). This is an interesting phenomenon, but the overall advance in understanding sex determination pathways is not so clear. The authors should engage with this paradox and discuss possible explanations for why *Sox8* is more important in this case of XX to testis sex reversal than it is in testis development in the XY gonad?

2) The authors claim that *Sox8* is expressed in the absence of *Rspo1* and *Sox9*. Based on Figure 1M-N, I would not consider this as being expressed, especially not when comparing between 1K to 1M (XY gonads). The induction of *Sox8* in XX gonads is also very weak but to similar extent in *Rspo1^KO^* and *Rspo1^KO^/Sox9^cKO^*. This is a main point that should be properly documented as the entire manuscript is based on *Sox8* being expressed even in the absence of *Sox9* (and SRY in XX). Have you tried to corroborate *Sox8* expression also with an antibody against *Sox8* or is there no specific antibody for *Sox8* that does not detect the other SoxE proteins? In light of this, are you sure that your *Sox9* antibody is specific and detects only *Sox9* and not *Sox8*/10? Have you considered looking at *Sox8* expression levels using qPCR or other quantitative methods (maybe RNA Scope)?

3) Overall the organization of the main figures and supporting figures make the experiments very hard to follow, even by someone from the field. It would be very helpful if the authors mention the figures and the frames of each figure in the text in the order they are presented in the figure (eg. discuss Figure 1A before 1B before 1C, and do not mention Figure 4 before describing Figures 1-3, etc). In some cases, frames need to be reorganized and some of the frames from the supporting figures might be better moved to the main figure if they are needed to understand the main figure. For example, the scheme at Figure 6—figure supplement 4 is very helpful and I would advise to incorporate it in the main text rather than the supplementary material. Please label the stage in all figures.

[Editors' note: further revisions were suggested prior to acceptance, as described below.]

Thank you for resubmitting your work entitled "*Sox8* and *Sox9* act redundantly for ovarian-to-testicular fate reprogramming in the absence of *Rspo1* in mouse sex reversal" for further consideration by *eLife*. Your revised article has been evaluated by Kathryn Cheah (Senior Editor) and a Reviewing Editor.

The manuscript has been improved but there are some remaining issues that need to be addressed before acceptance, as outlined below.

Summary:

This work follows on previous analyses by this lab and others, showing that loss of *Rspo1* or *Ctnnb1* in the context of loss of *Sox9* still resulted in testis development in XX animals, suggesting that another gene was responsible for driving testis development. In this manuscript, the authors use triple mutants to show that in the absence of the female pathway driver, *Rspo1*, XX gonads form testes if either *Sox8* or *Sox9* is present, but not if both are absent. These results show that in these circumstances, *Sox8* is a more important player than was previously appreciated.

Overall, the authors have done a nice job of responding to the previous round of review, and the revisions have clarified and improved the manuscript. A few technical issues remain that require the authors' attention. To point these out and make things easy, I have attached a marked-up version of the manuscript to this letter.

Revisions:

Please modify genotypes so that they are written correctly.

Please adjust lettering on figures or modify text so that all panels are described in order.

Please modify the English in a few places (marked).

---

## [Author Response]

Essential revisions:1) This analysis was an enormous amount of work, and the results are surprising: Since Sox8 is not critical to establish the testis pathway in the XY gonad, it is interesting that it is important to shift the XX gonad to the testis pathway. The authors primarily describe the change in expression of markers of the male and female pathways, and the physiological consequences on secondary sex characteristics, but do not get at why Sox8 is more important to drive testis fate in the XX gonad than it is to drive testis fate in the XY gonad (in mice). This is an interesting phenomenon, but the overall advance in understanding sex determination pathways is not so clear. The authors should engage with this paradox and discuss possible explanations for why Sox8 is more important in this case of XX to testis sex reversal than it is in testis development in the XY gonad?

This paradox between testicular differentiation in XY gonads and XX sex reversal is further discussed in the last paragraph of the Discussion:

“In mice, *Sox8* is dispensable for Sertoli cell differentiation (Sock et al., 2001) and an inefficient or late deletion of *Sox9* leads to XY sex reversal only if there is additional deletion of *Sox8* (Lavery et al., 2011, Chaboissier et al., 2004, Barrionuevo et al., 2009). This suggests that *Sox8* reinforces *Sox9* during testis differentiation. […] Thus, although *Sox8* is dispensable for testicular differentiation in wildtype mice, our current study demonstrates that *Sox8* is essential for testicular differentiation in sex reversal conditions.”

2) The authors claim that Sox8 is expressed in the absence of Rspo1 and Sox9. Based on Figure 1M-N, I would not consider this as being expressed, especially not when comparing between 1K to 1M (XY gonads). The induction of Sox8 in XX gonads is also very weak but to similar extent in Rspo1^KO^ and Rspo1^KO^/Sox9^cKO^. This is a main point that should be properly documented as the entire manuscript is based on Sox8 being expressed even in the absence of Sox9 (and SRY in XX). Have you tried to corroborate Sox8 expression also with an antibody against Sox8 or is there no specific antibody for Sox8 that does not detect the other SoxE proteins? In light of this, are you sure that your Sox9 antibody is specific and detects only Sox9 and not Sox8/10? Have you considered looking at Sox8 expression levels using qPCR or other quantitative methods (maybe RNA Scope)?

To get a better insight on the expression of *Sox8* in *Rspo1^KO^* and *Rspo1^KO^Sox9^cKO^*gonads, we obtained an anti-SOX8 antibody from Elisabeth Sock/Michael Wegner (Stolt et al., 2005) and performed a new pattern of expression analyses for SOX8 at E17.5 and P10 (see Figure 1B). The beginning of granulosa-to-Sertoli cell transdifferentiation occurs at E17.5, at which point we detected some SOX8-positive cells in XX *Rspo1^KO^* and XY/XX *Rspo1^KO^ Sox9^cKO^* gonads. At P10, there is a robust expression of SOX8 in these genotypes. This new immunostaining corroborates our previous work showing *Sox8* mRNA in XY/XX *Rspo1^KO^Sox9^cKO^* gonads (Lavery et al., 2012).

Regarding cross-reaction of anti-SOX9 (Σ, HPA001758) with other SOX8/10. To study the specificity of anti-SOX9, we performed immunodetection analyses on adrenals near birth (E18.5), when SOX10 is expressed in the medulla region (Reiprich et al., 2008). The absence of SOX9 expression in the adrenal indicates that the anti-SOX9 antibody does not cross-react with SOX10 antigen (see Author response image 1).

For cross-reactivity with SOX8, the absence of SOX9 immunodetection on XY *Rspo1^KO^ Sox9^cKO^* gonads (see Author response image 1) demonstrates that this anti-SOX9 antibody does not cross react with SOX8 antigen. As we previously demonstrated, these gonads express SOX8 (see Figure 1Bf and (Lavery et al., 2012)), but not SOX9 (genetic deletion of *Sox9^fl/fl^*by *Sf1:Cre^Tg/+^*).

Together, these data shows that this anti-SOX9 antibody is specific for SOX9 antigen.

**Author response image 1. respfig1:** Left panel – Immunostaining analysis using anti-SOX9 antibody on adrenal cross-sections at E18.5. Nuclear SOX9 is readily detectable in the ureter tips of the kidney, but not in the medulla of the adrenal gland. This confirmed that this antibody does not cross-react with SOX10. Middle and right panel – Immunostaining analyses using anti-SOX9 antibody on testes at P10. Anti-SOX9 antibodies reveal expression of SOX9 in XY *Rspo1*^KO^ gonads, but not in XY *Rspo1^KO^Sox9^cKO^* gonads, which we have demonstrated express SOX8. Thus, this anti-SOX9 antibody does not cross react with SOX8 antigen.

3) Overall the organization of the main figures and supporting figures make the experiments very hard to follow, even by someone from the field. It would be very helpful if the authors mention the figures and the frames of each figure in the text in the order they are presented in the figure (eg. discuss Figure 1A before 1B before 1C, and do not mention Figure 4 before describing Figures 1-3, etc). In some cases, frames need to be reorganized and some of the frames from the supporting figures might be better moved to the main figure if they are needed to understand the main figure. For example, the scheme at Figure 6—figure supplement 4 is very helpful and I would advise to incorporate it in the main text rather than the supplementary material. Please label the stage in all figures.

For all figures, we ensured that the stage of interest is clearly labeled. We also reorganized the figures/panels to follow the general order they are discussed in the text.

Previously, for Figure 1, XY and XX genotypes were alternated (XY, XX, XY, XX, etc.). To simplify this, we grouped XY genotypes and separated them from XX genotypes.

Previously, for Figures 2 and 5 (formerly Figure 6), we discussed figure panels from the supplemental figure before the main figure. To rectify this, we moved external genitalia panels to the main figure. This way, we can mention the main figure first and refer to the supplemental data as needed therein after.

Previously, the quantifications for immunostainings served as main figure (previously Figure 4) and the graphic summary (previously Figure 6—figure supplement 4) was presented as supplemental data. Now, the quantifications are supplemental data (Figure 3—figure supplement 2) and the graphic summary is a main figure (Figure 6).

The quantifications are now shown as a supplemental figure (Figure 3—figure supplement 2) and instead the former Figure 6—figure supplement 4 (graphic summary) has become a main Figure (Figure 6). The stages have been labeled in all figures. This way, the graphic summary genotypes is more accessible to the reader.

[Editors' note: further revisions were suggested prior to acceptance, as described below.]

Revisions:Please modify genotypes so that they are written correctly.

The inappropriately formatted genotypes have been rectified.

Please adjust lettering on figures or modify text so that all panels are described in order.

We now introduce the figures in the first sentence of their paragraph. This allowed us to keep the panels in alphabetical order without changes to the figure itself. We thank you for providing options to correct this.

Please modify the English in a few places (marked).

The spelling errors have been corrected.